# SWGA: A Distributed Hyperparameter Search Method for Time Series Prediction Models

## Abstract

We propose a distributed hyperparameter search method for time series prediction models named SWGA (Sliding Window Genetic Algorithm). Compared to current genetic algorithms for hyperparameter search, our method has three major advantages: (i) It adopts a configurable sliding window mechanism to effectively combat overfitting from distribution shifts inherent in time series data. (ii) It introduces a warm-up stage using Bayesian optimization-based methods to generate a good initial population. (iii) It supports distributed hyperparameter search across multi-node computing clusters, enhancing both scalability and efficiency. To demonstrate SWGA's efficacy, we conduct hyperparameter search experiments on time series datasets from various domains. The experiment results show that our method consistently finds a hyperparameter configuration that achieves better performance on out-of-sample time series data compared to the traditional genetic algorithm. On average, it reduces the out-of-sample loss by about 56.1%.

## 1 Introduction

In the realms of machine learning and deep learning, hyperparameter tuning stands as a cornerstone to effective model training. It is important to tune the hyperparameters of the model on a validation dataset. First, this adjustment helps minimize the risk of model overfitting to the training data, which often severely degrades out-of-sample performance. Second, by fine-tuning hyperparameters on a validation set with a distribution similar to the training data, the model can achieve better performance on out-of-sample data with a matching distribution. Lastly, since many hyperparameters pertain to the model architecture and computational efficiency, optimal configurations can enhance model efficiency. Consequently, researchers widely adopt hyperparameter tuning across various machine learning and deep learning domains Vaswani et al. (2017); Dosovitskiy et al. (2020); Zhang & Yan (2022); Liu et al. (2021); Zhou et al. (2021); Devlin et al. (2018); Arik & Pfister (2021); Huang et al. (2020).

Time series prediction remains a crucial endeavor in various sectors. Domains such as energy Hong et al. (2020); Nti et al. (2020); Reneau et al. (2023), finance Fischer & Krauss (2018), house pricing Xu & Zhang (2021), and medical treatment Prakarsha & Sharma (2022) heavily rely on predicting future time series values based on historical data. With the swift advances in machine learning, optimizing performance on unseen data demands rigorous hyperparameter search. Time series data, characterized by temporal dependencies and non-stationarity, poses unique challenges. Temporal dependencies mandate models to discern patterns evolving with time, while non-stationarity implies fluctuating statistical properties, leading to potential distribution shifts Kim et al. (2021); Fan et al. (2023) and possible model overfitting Roelofs et al. (2019).

Traditional general-purpose hyperparameter search algorithms do not take into account the domain knowledge of the time series prediction problem by design. Specifically, many time series prediction models suffer from non-stationary time series and the temporal distribution shift in the dataset is a long-lasting problem Du et al. (2021). In this work, we propose

a hyperparameter search process that caters for temporal distribution shifts in time series data.

Addressing these challenges, we present the Sliding Window Genetic Algorithm (SWGA), a pioneering method tailored for hyperparameter search in time series prediction models. SWGA offers three innovations: a sliding window technique to mitigate overfitting due to time series distribution shifts, a warm-up phase that utilizes Bayesian optimization for crafting a solid initial population, and inherent compatibility with distributed computation across multi-node clusters.

This paper delves into SWGA's underlying methodology and assesses its efficacy across diverse time series datasets, consistently demonstrating its edge over conventional genetic algorithms in identifying optimal hyperparameters for out-of-sample time series predictions.

There are four major contributions of this paper:

- We introduce a warm-up stage using a lightweight TPE method, enhancing the initialization of the initial population. Compared to the random initialization in traditional genetic algorithms, this approach offers a more promising starting point for subsequent iterations, ultimately guiding the algorithm towards optimal convergence.
- We unveil a configurable sliding window mechanism for hyperparameter search tailored for time series datasets, bolstering the search's resilience against distribution shifts in time series data.
- We demonstrate an effective way to incorporate the consideration of the distribution shift in time series into the hyperparameter search process to create a domain-knowledge-enhanced hyperparameter search method that is better than its general-purpose counterpart. Using genetic algorithm (GA) as an example in the experiments, we demonstrated that our proposed way (warm-up and sliding window) can greatly enhance the base method, GA, into SWGA, a method that gives much better out-of-sample results for time series prediction models.
- Our algorithm seamlessly integrates with the Ray distributed computation framework Moritz et al. (2018), making it adaptable to a wide range of parallelism scenarios.

We structure the rest of the paper as follows: Section 2 reviews related works. Section 3 provides the necessary background. Section 4 elaborates on the SWGA methodology. Section 5 outlines our experimental design, datasets, and results. Section 6 is the conclusion of the paper.

## 2 RELATED WORK

In this section, we discuss the relevant literature on hyperparameter tuning methods for time series prediction, covering traditional optimization techniques, distributed computing approaches, and evolutionary algorithms.

Researchers widely use traditional optimization techniques, such as grid search and random search Bergstra & Bengio (2012), for hyperparameter tuning in time series prediction models. Although these methods are conceptually simple, they come with high computational costs and inefficient exploration of large hyperparameter search spaces. Bayesian optimization methods, which gained popularity due to their ability to model the performance landscape and guide the search towards promising regions of the hyperparameter space Snoek et al. (2012), still demand substantial computational resources for large-scale time series prediction problems.

To tackle the computational challenges associated with hyperparameter tuning, researchers propose distributed computing approaches. Some examples include Population-based Training (PBT) Jaderberg et al. (2017), Asynchronous Successive Halving Algorithm (ASHA) Li et al. (2020), and Hyperband Li et al. (2017). These methods exploit parallelism to accelerate the search and find success in various machine learning tasks. But, their full applicability to time series prediction problems requires further study, and adaptations may be necessary

to handle the unique challenges of time series data, such as non-stationarity and temporal dependencies.

Researchers employ evolutionary algorithms, such as Genetic Algorithms (GAs), for hyperparameter optimization in various machine learning tasks Alibrahim & Ludwig (2021) Elgeldawi et al. (2021). GAs exhibit several attractive properties, such as global search capabilities and robustness to local optima, making them suitable for complex optimization problems. The literature contains several distributed GA variants, including Distributed Genetic Algorithm (DGA) Belding (1995), Island Model Genetic Algorithm Whitley et al. (1999), and Master-Slave Genetic Algorithm Cantu-Paz & Goldberg (2000). While these methods apply to a wide range of optimization problems, their application to time series prediction tasks remains limited.

K-fold cross-validation Kohavi et al. (1995) is a popular technique used for model evaluation and hyperparameter tuning in machine learning. This method involves partitioning the dataset into K equally sized folds, where each fold serves as a validation set exactly once, while the remaining K-1 folds are used for training the model. By averaging the performance metrics across all K iterations, K-fold cross-validation provides a more robust and reliable estimate of the model's generalization performance compared to a single train-test split. This approach is particularly useful in scenarios where the dataset size is limited, as it maximizes the usage of available data for both training and evaluation. Moreover, K-fold cross-validation effectively reduces the risk of overfitting and helps to identify a model that generalizes well to new, unseen data. Our algorithm may look similar to K-fold cross-validation, but they are very different.

Ray Moritz et al. (2018) is a distributed computing framework that supports various distributed computing infrastructures. We integrate it into our algorithm implementation to enable the distributed hyperparameter search capability.

## 3 BACKGROUND

**Time series prediction** Suppose that we have a multivariate time series with $N$ variates. It is also a set of $N$ univariate time series $\{z_{1:T_0}^i\}_{i=1}^N$. There are in total $T_0$ time steps. The prediction target is the next $\tau$ time steps $\{z_{T_0+1:T_0+\tau}^i\}_{i=1}^N$. We are trying to model:

$$p(z_{T_0+1:T_0+\tau}^i|\{z_{1:T_0}^i\}_{i=1}^N; \Phi) = \prod_{i=1}^{\tau} p(z_{T_i}|\{z_{1:T_0}^i\}_{i=1}^N; \Phi)$$

In this conditional distribution, $\Phi$ is the parameter of the prediction model.

**Hyperparameter search** Consider a machine learning model $M$ characterized by a set of hyperparameters $H = \{h_1, h_2, ..., h_n\}$. Each hyperparameter $h_i$ has a domain $d(h_i)$ from which a value can be selected. The goal of hyperparameter search is to find a configuration $C = \{c_1, c_2, ..., c_n\}$, where each $c_i \in d(h_i)$, that optimizes the performance of the model $M$ on a given dataset. This can be mathematically formulated as:

$$C^* = \arg \min_{C \in d(H)} L(M(H = C), D) \tag{1}$$

Here, $L$ represents a loss function that quantifies the discrepancy between the predictions of the model $M$ with hyperparameters set to $C$ and the true values in the dataset $D$. The aim is to find the hyperparameter configuration $C^*$ that minimizes this loss.

**Distribution shift** Time series prediction models often suffer from non-stationarity from the time series data. The distribution in these data shifts along the time direction. To mitigate distribution shift, people usually use domain generalization (Li et al. (2018); Muandet et al. (2013); Wang et al. (2022)) and domain adaptation (Tzeng et al. (2017); Ganin et al. (2016); Wang et al. (2018)). Domain generalization focuses on learning from the source domain and hopes to generalize well on the target domain while domain adaptation is to

reduce the distribution distance between the source and target domain. They both have the goal to bridge the distributions of source and target domains. However, our method is different from these methods in the sense that we address the distribution shift from the hyperparameter search perspective.

**Tree-structured Parzen Estimator**   The Tree-structured Parzen Estimator (TPE) is a prominent method for hyperparameter optimization. TPE models the joint distribution $p(x, y)$ of the hyperparameters $x$ and the objective function $y$. In contrast to other optimization techniques that model $p(y|x)$ and then invert this relationship, TPE models $p(x|y)$ and $p(y)$ directly. TPE divides the hyperparameters into two sets depending on the observed $y$ values and then generates new candidate hyperparameters from a distribution that favors the promising set. In doing so, TPE provides a more flexible way of exploring the hyperparameter space, especially when the distribution of hyperparameters is non-uniform. However, TPE can be computationally expensive as the number of hyperparameters grows and sensitive to the choice of the threshold that separates the two sets of hyperparameters. Besides, TPE runs in a sequential manner. It is hard to run in parallel and utilize the modern multi-node distributed computing clusters.

## 4 METHODOLOGY

To initialize the first population, rather than using random generation, we use a Tree-structured Parzen Estimator (TPE) to repeatedly run a small number of trials to generate the initial population. We call this process the warm-up stage and it provides a better starting population for the genetic algorithm. To make the hyperparameter search more robust to the time series's distribution shift problem and prevent the algorithm from overfitting the validation set, we create this configurable sliding window mechanism when conducting the genetic algorithm. We first split a dataset into the training set, validation set, and testing set according to a fixed ratio. Then, we evenly split the training set into multiple chunks of the same size. Then, we split the validation set into the same number of chunks. The hyperparameter search process goes as follows. First, we define a window of a length of a fixed number of chunks. The window starts from the earliest chunk and slides one chunk after each iteration of SWGA along the time direction. Starting from the population of the first generation, at each iteration, SWGA trains the model with each individual config in the population only on the data within the fixed-length window and does the model validation on the data chunk right after the window. Then, the window slides along the time dimension using a fixed stride of one data chunk. The size of each data chunk and the size of the window are both configurable. Figure 1 shows an example of how SWGA works on a 3-year time series dataset.

In detail, The entire process of SWGA, also shown by Algorithm 1 - 5, is as follows. First, it splits the datasets into the training set, validation set, and testing set. In the warm-up stage, the TPE algorithm runs a small number of trials on the training set and the validations set to produce one hyperparameter config. This process repeats several times until it generates enough individuals for the initial population. Then, the genetic algorithm process starts. In each iteration, it first evaluates each individual in the population and sorts them according to the ascending fitness value. The fitness value is the trained model's Root Mean Square Error (RMSE) or Mean Absolute Error (MAE) on the validation data. Then, based on the ranking of each individual in the population regarding the fitness value, the algorithm generates a new population for the next iteration. The population generation process is as follows. It first creates a set of parents from the top k individuals (low fitness values) and from the tail 2 individuals of the population. Then, it applies the crossover operator and mutation operator to the parents to generate the offsprings. The offsprings and the top k individuals together become the next generation of the population. Lastly, after the final iteration, the top individual in the population becomes the final winner. SWGA then uses this configuration to train a model on the original training set and report the RMSE or MAE on the testing set.

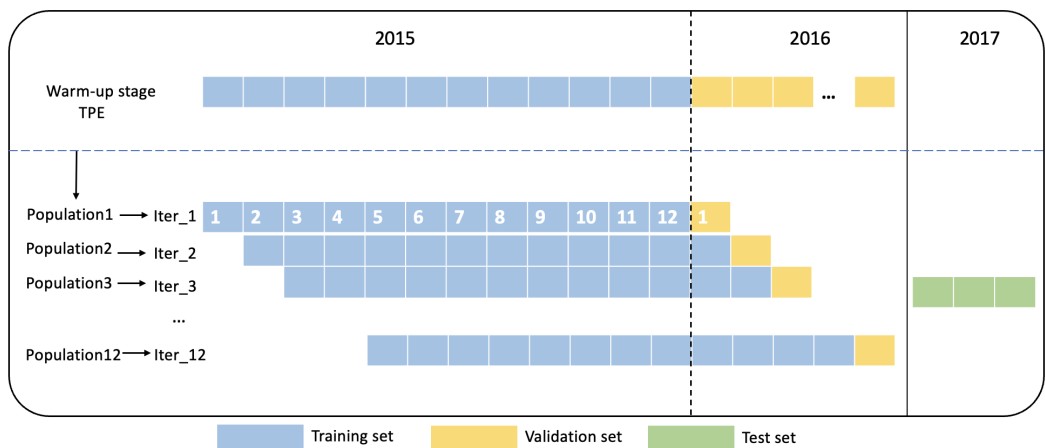

Figure 1: A demonstration of SWGA. From top to bottom, they are the different stages of the tuning process. In this specific example, the training set is the history time series for 2015 and the validation set is the history time series for 2016. They are both divided into 12 chunks respectively. The test set is the history time series for 2017. In each iteration, there is a sliding window including in total 13 chunks with 12 as training set and 1 as validation set. It generates the population on this window for the next iteration. The best configuration from the final generation is used to obtain the evaluation metrics on the test set.

**SWGA variant**  We also consider a variant of the SWGA. In each iteration, rather than only using one data chunk right after the training window to validate the model, we use all the data chunks in the validation set that is not in the window. However, this variant is much slower and the experiment results show that it does not provide a better performance. Thus, we do not use this variant to run the experiments in the experiment section.

Since genetic algorithms are natively parallelization-friendly, we also integrate SWGA with Ray compute framework to support parallelized hyperparameter search on various computation infrastructures including single-node multi-core and multi-node multi-core.

---

**Algorithm 1** SWGA

---

1: Raw dataset separates into $trn\_set, val\_set, tst\_set$
2: Initialize $population$ as an empty list
3: Initialize $fitnesses$ as an empty list
4: **for** $i = 1, 2, \ldots, K$ **do**
5:   **if** $i == 1$ **then**
6:     $population_0 =$
$$\text{WarmUpStage}(trn\_set, val\_set)$$
7:     $trn\_set$ splits into $N$ chunks
8:     $val\_set$ splits into $N$ chunks
9:   **end if**
10:   $trn\_set_i, val\_set_i = \text{GetDataset}(i, trn\_set, val\_set)$
11:   $population_i =$
$$\text{GenNextPop}(population_{i-1}, fitnesses_{i-1})$$
12:   **for** each individual configuration $h$ in $population_i$ **do**
13:     Evaluate the fitness of $h$ and add to $fitnesses_i$
14:   **end for**
15: **end for**
16: **return**  the config with the best fitness in $fitnesses_K$

---

---

**Algorithm 2** GetDataset($i$, $trn\_set$, $val\_set$)

---

1: $trn =$
    concatenate($trn\_set[chunk_i, \ldots, chunk_{i+N-1}]$)
2: $val = val\_set[chunk_{i+N}]$
3: **return** $trn, val$

---

---

**Algorithm 3** CrossoverMutate($parent1$, $parent2$)

---

1: $crossover\_rate = 0.7$
2: $mutate\_rate = 0.2$
3: **for** each hyperparameter key $c$ in the config space **do**
4:   $child[c] = $ RandomChoice($parent1[c]$, $parent2[c]$)
5:   $child[c] = $ RandomChoice($child[c]$, $default\_config[c]$) # mutation
6: **end for**
7: **return** $child$

---

---

**Algorithm 4** WarmUpStage($trn\_set, val\_set$)

---

1: Initialize $init\_pop$ as an empty list
2: **while** size_of($init\_pop$) < POPULATION_SIZE **do**
3:   best_config = TPE(num_trials=10)
4:   Add best_config to $init\_pop$
5: **end while**
6: **return** $init\_pop$

---

---

**Algorithm 5** GenNextPop($population_{i-1}$, $fitnesses_{i-1}$)

---

1: Get $topk\_selection$ from $population_{i-1}$ according to fitness in $fitnesses_{i-1}$
2: Get $tail2\_selection$ from $population_{i-1}$ according to fitness in $fitnesses_{i-1}$
3: Initialize $parent\_pairs$ as empty list
4: Initialize $next\_pop$ as empty list
5: **for** $i = 1, 3, 5, \ldots$, k-1 (two elements each time) **do**
6:   Append ($topk\_selection[i]$, $topk\_selection[i+1]$) to $parent\_pairs$
7: **end for**
8: Append ($tail2\_selection[0]$, $tail2\_selection[1]$) to $parent\_pairs$
9: **for** each $parent\_pair$ in $parent\_pairs$ **do**
10:   $child = $ CrossoverMutate($parent\_pair$)
11:   Add $child$ to $next\_pop$
12: **end for**
13: **return** $next\_pop \cup topk\_selection$

---

## 5 EXPERIMENT

To demonstrate the efficacy of our methodology, we conduct two tasks in our experiments.

**Task 1**  In this task, we focus on showcasing the effectiveness of our proposed methodology by ablation studies. We use SWGA, GA and SWGA* (SWGA without the warm-up stage) to search for hyperparameters for 5 common time series prediction model architectures respectively. Then, we compare the out-of-sample prediction performance of these models. Through this task, we show that both our proposed warm-up stage and the sliding window mechanism are effective and our proposed SWGA method indeed has a performance gain compared to the base GA method. The results are in Table 1 and 2.

**Task 2**  In this task, we demonstrate SWGA's values in real applications. We use SWGA to search for hyperparameters for three latest SOTA time series prediction models in the literature including iTransformer Liu et al. (2023), DLinear Zeng et al. (2023) and PatchTST Nie et al. (2022). We show these models' immediate improvements regarding the out-of-

sample prediction performance on the same long-term forecasting task, training, and testing dataset as the original setups in the literature.

**Experiment Configurations**  We conduct all the experiments on a Ray cluster node with 48 CPU cores and 8 Nvidia RTX 2080Ti GPUs. For Task 1, we use SWGA to do a hyperparameter search on 4 different prediction models on 10 multivariate time series datasets from different domains. For each dataset, we first split the dataset using the 8:1:1 ratio into the training set, validation set, and testing set. Then, we split the training set and validation set respectively into 12 trunks. We use seven historical timesteps to predict one timestep ahead. Each experiment repeats 5 times and we report the mean RMSE and mean MAE. Since SWGA has the sliding window mechanism that increases the number of trials on different hyperparameter configurations, to ensure that there is a fair comparison, we make sure all the compared methods including the baseline have the same total number of trials in the hyperparameter tuning process. To obtain the RMSE and the MAE, we first use the hyperparameters that the hyperparameter search method finds to initialize the model. Then, we train the model on the training set and test the model on the test set. We report the model's RMSE and MAE on the test set. For Task 2, we ensure that all models have the same settings as that in Table 1 of the iTransformer Liu et al. (2023) paper. The only difference is that we use SWGA to do hyperparameter search. The hyperparameter search space we use is in A.1

Table 1: Comparison of RMSEs between SWGA and GA for different models and datasets. SWGA achieves the best results on most of the models and datasets. SWGA* represents the version of SWGA that does not use TPE to generate the initial population. Instead, it uses the random sampling method.

| Model | Method | Dataset | | | | | | | | | |
|---|---|---|---|---|---|---|---|---|---|---|---|
| | | Beijing PM2.5 | SML2010 | Appliance Energy | Individual Household Electricity | Exchange | ETTh1 | ETTh2 | ETTm1 | ETTm2 | Traffic |
| Catboost | SWGA | **0.071** | **0.055** | **0.069** | 0.032 | **0.015** | **0.034** | **0.056** | **0.027** | **0.015** | **0.015** |
| | SWGA* | 0.073 | 0.056 | 0.074 | **0.022** | **0.015** | 0.073 | **0.056** | 0.074 | 0.022 | 0.039 |
| | GA | 0.081 | 0.137 | 0.070 | 0.078 | 0.080 | 0.062 | 0.117 | 0.038 | 0.065 | 0.067 |
| LightGBM | SWGA | **0.079** | 0.159 | **0.068** | **0.062** | **0.051** | 0.072 | 0.095 | **0.051** | **0.052** | **0.051** |
| | SWGA* | 0.088 | 0.164 | 0.078 | 0.065 | **0.051** | **0.071** | **0.093** | 0.070 | 0.085 | 0.054 |
| | GA | 0.080 | **0.137** | 0.078 | 0.078 | 0.078 | 0.082 | 0.108 | 0.073 | 0.096 | 0.067 |
| XGBoost | SWGA | **0.087** | 0.136 | **0.108** | 0.070 | **0.049** | **0.080** | **0.100** | 0.108 | **0.050** | 0.078 |
| | SWGA* | 0.091 | 0.159 | 0.147 | **0.052** | 0.052 | 0.091 | 0.159 | 0.147 | 0.052 | **0.052** |
| | GA | 0.414 | **0.121** | 0.440 | 0.426 | 0.208 | 0.081 | 0.145 | **0.075** | 0.132 | 0.376 |
| LSTM | SWGA | **0.087** | **0.265** | **0.092** | **0.065** | **0.013** | **0.087** | **0.265** | **0.092** | **0.065** | 0.024 |
| | SWGA* | 0.198 | 0.349 | 0.180 | 0.568 | 0.017 | 0.198 | 0.369 | 0.180 | 0.568 | **0.023** |
| | GA | 0.168 | 0.636 | 0.197 | 0.176 | 0.260 | 0.737 | 0.649 | 0.593 | 0.656 | 0.110 |
| Transformer | SWGA | **0.071** | **0.121** | **0.071** | **0.058** | 0.109 | **0.056** | **0.040** | **0.078** | 0.109 | **0.042** |
| | SWGA* | 0.072 | 0.197 | 0.089 | 0.084 | **0.095** | 0.118 | 0.181 | 0.102 | 0.143 | 0.084 |
| | GA | 0.608 | 0.879 | 0.707 | 0.911 | 0.730 | 1.312 | 0.990 | 0.732 | 1.142 | 0.578 |

**Datasets**  In the experiments, we use ten real-world datasets: (i) Beijing PM2.5: This is an hourly multivariate time series dataset ranging from 2010 to 2014. It has (ii) SML2010: A month of home monitoring multivariate time series data of resolution of 15 minutes. (iii) Appliance Energy: Four months of energy use multivariate time series dataset of 10-minute resolution. (iv) Individual household electricity: Four years electricity use multivariate time series dataset of 1-minute resolution. (v) Exchange: It is a multivariate dataset including daily exchange rates in eight different countries from 1990 to 2010. (vi) ETT (Electricity Transformer Temperature) datasets are multivariate time series. There are two collection sources of them with labels 1 and 2. There are two collection resolutions that are 1 hour and 15 minutes. So, there are four specific datasets in this category: ETTh1, ETTh2, ETTm1, and ETTm2. (vii) Traffic: A multivariate dataset recording the hourly road occupancy rates from various sensors on freeways in San Francisco from 2016 to 2018.

Table 2: Comparison of MAEs between SWGA and GA for different models and datasets. SWGA achieves the best results on most of the models and datasets.

| Model | Method | Dataset | | | | | | | | | |
|---|---|---|---|---|---|---|---|---|---|---|---|
| | | Beijing PM2.5 | SML2010 | Appliance Energy | Individual Household Electricity | Exchange | ETTh1 | ETTh2 | ETTm1 | ETTm2 | Traffic |
| Catboost | SWGA | **0.018** | **0.039** | **0.026** | **0.015** | 0.045 | **0.058** | **0.023** | 0.053 | **0.023** | 0.016 |
| | SWGA* | 0.034 | 0.116 | 0.027 | 0.016 | 0.054 | 0.143 | 0.180 | **0.024** | 0.132 | **0.014** |
| | GA | 0.062 | 0.117 | 0.038 | 0.065 | **0.028** | 0.070 | 0.092 | 0.060 | 0.082 | 0.058 |
| LightGBM | SWGA | **0.067** | 0.120 | 0.050 | 0.063 | **0.051** | 0.068 | **0.077** | 0.045 | **0.057** | **0.051** |
| | SWGA* | 0.072 | 0.114 | 0.051 | **0.052** | 0.052 | **0.067** | 0.082 | 0.047 | 0.058 | 0.053 |
| | GA | 0.077 | **0.112** | **0.045** | 0.068 | 0.074 | 0.071 | 0.092 | 0.056 | 0.089 | 0.057 |
| XGBoost | SWGA | 0.089 | 0.106 | 0.313 | **0.022** | **0.042** | 0.048 | 0.045 | 0.045 | 0.042 | **0.034** |
| | SWGA* | **0.080** | 0.110 | **0.108** | 0.275 | 0.059 | 0.049 | 0.063 | 0.049 | 0.060 | 0.199 |
| | GA | 0.410 | **0.103** | 0.418 | 0.416 | 0.074 | 0.067 | 0.114 | 0.067 | 0.108 | 0.371 |
| LSTM | SWGA | **0.018** | **0.040** | 0.029 | **0.016** | 0.014 | 0.048 | 0.021 | 0.038 | 0.015 | **0.016** |
| | SWGA* | 0.036 | 0.059 | **0.025** | 0.017 | 0.014 | 0.197 | 0.080 | 0.128 | 0.119 | 0.024 |
| | GA | 0.236 | 0.665 | 0.217 | 0.089 | 0.288 | 0.491 | 0.569 | 0.719 | 0.583 | 0.152 |
| Transformer | SWGA | **0.069** | **0.109** | 0.056 | **0.053** | **0.093** | **0.083** | **0.105** | **0.061** | **0.072** | **0.060** |
| | SWGA* | 0.076 | 0.170 | **0.051** | 0.065 | 0.125 | 0.163 | 0.121 | 0.082 | 0.084 | 0.156 |
| | GA | 0.532 | 0.612 | 0.790 | 0.434 | 0.770 | 0.790 | 0.811 | 1.210 | 1.229 | 0.993 |

**Out-of-sample performance (task 1)** As we can see from Table 1, SWGA consistently outperforms the traditional genetic algorithm on most of the datasets and different models. On average, SWGA reduces the RMSE on the out-of-sample testing set by 54.6% compared to GA. SWGA* is the SWGA without the warm-up stage. Instead, SWGA* uses the random sampling method to generate the initial population as the traditional genetic algorithm. On average, SWGA* reduces the RMSE on the out-of-sample testing set by 34.0% compared to GA. By comparing the results of SWGA* and the results of GA, we can know that the configurable sliding window mechanism indeed brings a significant reduction to the out-of-sample RMSE. By comparing the results of SWGA and SWGA*, we can see that the warm-up stage contributes additional reduction to the RMSE on top of the sliding window's contribution in most cases.

Table 2's results are consistent with Table 1. SWGA has the lowest MAE on most of the datasets. On average, SWGA has about a 57.6% reduction compared to the MAE of GA. SWGA* reduces the MAE by about 42.6% compared to GA. In both the MAE and RMSE metrics, SWGA yields a significant improvement over GA.

Besides, SWGA shows a consistent advantage in both Table 1 2 across various kinds of popular time series prediction model architectures including tree models (Catboost, LightGBM, XGBoost), recurrent models (LSTM), and attention-based models (Transformer). This further demonstrates SWGA's advantage and application value.

Table 3: The improvement of results on testing dataset of three SOTA time series forecasting models by using SWGA to search for a better set of hyperparameters. We calculate and show the average percentage of the reduction of the mean square error (MSE) after using SWGA to do hyperparameter search. We can see that Each of them has a considerable amount of free improvement without any change to their dataset and model architecture.

| Model | Dataset | | | | | |
|---|---|---|---|---|---|---|
| | ETTh1 | Weather | ETTm1 | Exchange | ETTh2 | ETTm2 |
| iTransformer | 1.92% | 1.90% | 2.20% | 1.14% | 0.80% | 4.00% |
| DLinear | 0.80% | 2.84% | 5.50% | 1.30% | 4.25% | 6.05% |
| PatchTST | 6.46% | 1.14% | 3.30% | 6.13% | 1.60% | 3.39% |

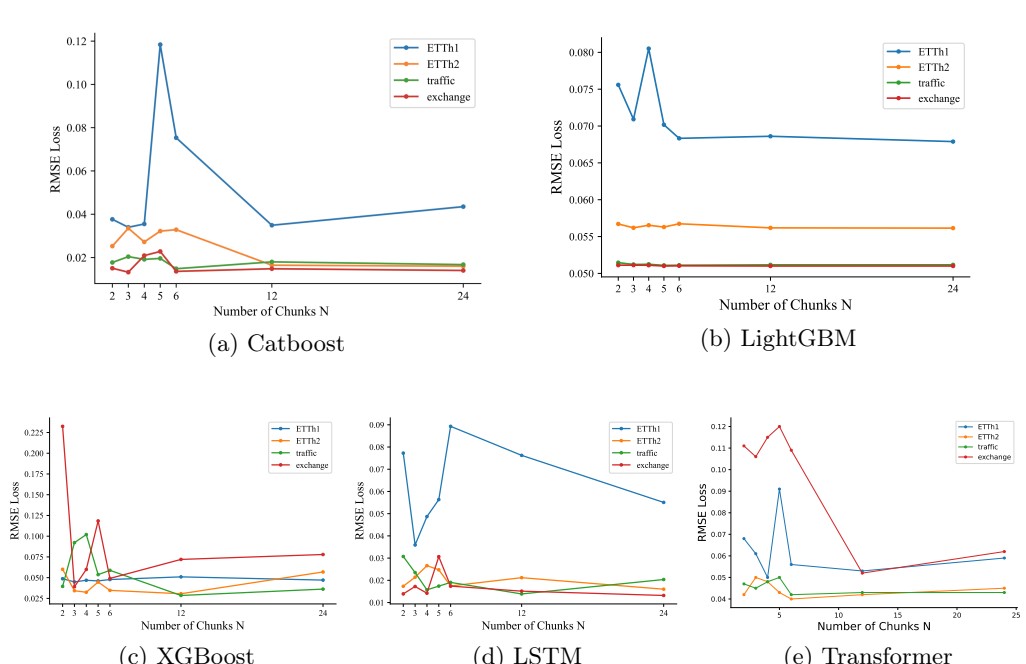

Figure 2: SWGA testing loss (RMSE) for different numbers of chunks (N).

**Improvement on latest SOTA time series forecasting models (task 2)**  As we can see from Table 3, by using SWGA to do hyperparameter search on three SOTA time series forecasting models, without other additional modifications, we immediately get an improvement as much as 6.46% of average reduction of MSE. This demonstrates that a considerable amount of additional testing performance of time series predictions models is achievable by using a good set of hyperparameters. Besides, it demonstrates our SWGA method's capability of gaining such addition testing performance on a wide range of existing SOTA models in an easy plug-and-play manner.

**Number of chunks**  The above experiments set the number of training chunks $N$ to 12 and it already produces a much better performance than the baseline GA. To investigate the effect of different $N$s on the out-of-sample testing loss, we conduct experiments adjusting the value of $N$ in SWGA. Figure 2 shows that different models and datasets have their own optimal $N$ values. For instance, for XGBoost, SWGA with $N = 6$ exhibits the best out-of-sample RMSE for all four of those datasets. The different effects from $N$ further prove that our sliding window mechanism is meaningful and necessary for time series data.

**Scalability**  To examine how varying the number of distributed computer nodes impacts optimisation time. We conduct experiments by adjusting the number of nodes in SWGA. All experiments in this section are conducted using the ETTh1 dataset. As depicted in Figure 3, we observe a reduction in optimization time with an increase in the number of nodes, and this relationship appeared nearly linear. The results demonstrate the good scalability and efficiency of our proposed framework.

## 6 CONCLUSION

We propose SWGA, a distributed genetic algorithm for hyperparameter search for time series data. Compared to a regular genetic-based algorithm that uses random initialization to initialize the initial population, we propose a warm-up stage that uses TPE with a small number of trials to generate the initial population to provide a better starting point. To combat the distribution shift challenge on time series datasets, we propose a configurable sliding window mechanism. Besides, SWGA natively supports parallelized hyperparameter

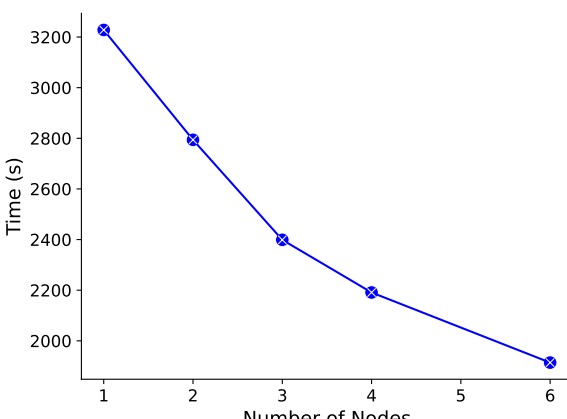

Figure 3: When the number of computation nodes increases, the optimization time decreases nearly linearly.

search on a Ray cluster. The experiment results on various models and time series datasets from different domains show that SWGA has a huge performance gain over the vanilla genetic algorithm. On average, there is a decrease of roughly 57.6% in the MAE and 54.6% in the RMSE when using SWGA in comparison to GA. Additionally, we also demonstrate the good scalability of SWGA.

**Boarder Impact.**   This paper presents work whose goal is to advance the field of Machine Learning. There are many potential societal consequences of our work, none which we feel must be specifically highlighted here.

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

# A   APPENDIX

## A.1   HYPERPARAMETER

The hyperparameters we tuned for each model are as follows.

Table 4: Hyperparameters For Models

| Model | Hyperparameter | Space |
|---|---|---|
| LSTM | learning_rate | [1e-6, 1e-1] |
| | num_layers | [4, 64] |
| | hidden_size | [4, 128] |
| | max_epochs | [5, 100] |
| | batch_size | {16, 32, 64, 128, 256, 512} |
| | dropout | [0.1, 0.5] |
| Catboost | learning_rate | {1e-2, 1e-3, 1e-4, 1e-5, 1e-6} |
| | iterations | [5, 80] |
| | depth | [4, 12] |
| | random_strength | [1, 8] |
| | l2_leaf_reg | [1e-3, 1e3] |
| | bagging_temperature | [0, 10] |
| Lightgbm | learning_rate | {1e-2, 1e-3, 1e-4, 1e-5, 1e-6} |
| | n_estimators | {5, 10, 20, 40, 80} |
| | max_depth | {4, 6, 8, 10]} |
| | lambda_l2 | {16, 32, 64, 128} |
| XGBoost | learning_rate | {1e-2, 1e-3, 1e-4, 1e-5, 1e-6} |
| | n_estimators | [5, 80] |
| | max_depth | [4, 12] |
| | reg_lambda | [1e-3, 1e3] |
| iTransformer | learning_rate | {1e-2, 1e-3, 1e-4, 1e-5, 1e-6} |
| | d_model | {32, 64, 96, 128, 160, 192, 224, 256 } |
| | encoder_layers | [1, 10] |
| DLinear | learning_rate | {1e-2, 1e-3, 1e-4, 1e-5, 1e-6} |
| | d_model | {32, 64, 96, 128, 160, 192, 224, 256 } |
| | encoder_layers | [1, 10] |
| PatchTST | learning_rate | {1e-2, 1e-3, 1e-4, 1e-5, 1e-6} |
| | d_model | {32, 64, 96, 128, 160, 192, 224, 256 } |
| | encoder_layers | [1, 10] |

## A.2   DATASET INFORMATION

We use 10 different multivariate time series datasets in the paper. They are all commonly used time series datasets. These datasets are from different domains, of different resolutions, and have different numbers of variates. We chose such diverse multivariate time series datasets to demonstrate our method's general efficacy. The following are some brief introductions and a table including the details of the datasets we used.

Beijing PM2.5[1] includes hourly multivariate data from 2010 to 2014. SML2010[2] is a month of home monitoring multivariate data of the resolution of 15 minutes. Appliance Energy[3] has 4 months of energy use data of 10-minute resolution. Individual Household Electricity[4]

---

[1]https://archive.ics.uci.edu/dataset/381/beijing+pm2+5+data
[2]https://archive.ics.uci.edu/dataset/274/sml2010
[3]https://archive.ics.uci.edu/dataset/374/appliances+energy+prediction
[4]https://archive.ics.uci.edu/dataset/235/
individual+household+electric+power+consumption

is 4 years of electricity use. The Exchange dataset[5] includes daily exchange rates in eight different countries from 1990 to 2016. The ETT(Electricity Transformer Temperature) dataset[6] datasets are multivariate time series. There are two collection sources of them with labels 1 and 2. There are two collection resolutions that are 1 hour and 15 minutes. So, there are four specific datasets in this category: ETTh1, ETTh2, ETTm1, and ETTm2. The Traffic dataset[7] consists of hourly road occupancy rates from California's Department of Transportation on San Francisco Bay area freeways. All datasets are split into training, validation, and test sets in an 8:1:1 ratio chronologically.

Table 5: Dataset Details

| Dataset | Number of Samples | Number of Variates |
|---|---|---|
| Beijing PM2.5 | 43824 | 13 |
| SML2010 | 4137 | 24 |
| AE | 19735 | 29 |
| IHE | 2075259 | 9 |
| Exchange | 7589 | 8 |
| ETTh1 | 17421 | 7 |
| ETTh2 | 17421 | 7 |
| ETTm1 | 69681 | 7 |
| ETTm2 | 69681 | 7 |
| Traffic | 17544 | 862 |

(AE: Appliance Energy, IHE: Individual Household Electricity)

### A.3 COMPUTATION HARDWARE AND SOFTWARE

All experiments are conducted on a cluster (except the distributed compute node experiment), where each node has 8 NVIDIA GEFORCE RTX 2080 Ti GPUs and 4 12-core Intel XEON Silver 4214 @ 2.20GHz. The total RAM is 790GB. The operating system is Ubuntu 18.04. The random seed we used was $\{1, 2, 5, 10, 24\}$. The major software and framework we used are PyTorch[8], scikit-learn[9], and Ray[10].

For the scalability experiments, the computing setup consists of computation nodes equipped with 16 Intel(R) Xeon(R) Gold 6230R CPUs and 1 A100 GPU each, with a combined RAM capacity of 1024G.

### A.4 OPTIMIZATION DYNAMICS

We conduct experiments to show the optimization dynamics of the baseline, GA algorithm on the four models on those four different datasets. Figure 4 and Figure 5 have three major takeaways: (i) For all four models, on most of the datasets, SWGA is able to reach a much lower out-of-sample testing loss compared to the baseline GA. (ii) SWGA's out-of-sample testing loss decreases in a smoother way while the baseline GA's loss optimization process is much more volatile bouncing up and down. This indicates that it is safer to use the tuned hyperparameter configuration from SWGA compared to that from the baseline GA where there is a higher chance that the tuned configuration is on the out-of-sample testing loss peak that bounces up from a previous local minimum. (iii) In some cases such as the XGBoost case, the out-of-sample loss from the base GA fails to decrease properly while the SWGA is able to.

---

[5]https://github.com/laiguokun/multivariate-time-series-data

[6]https://github.com/zhouhaoyi/ETDataset

[7]http://pems.dot.ca.gov

[8]https://pytorch.org/

[9]https://scikit-learn.org/stable/

[10]https://www.ray.io/

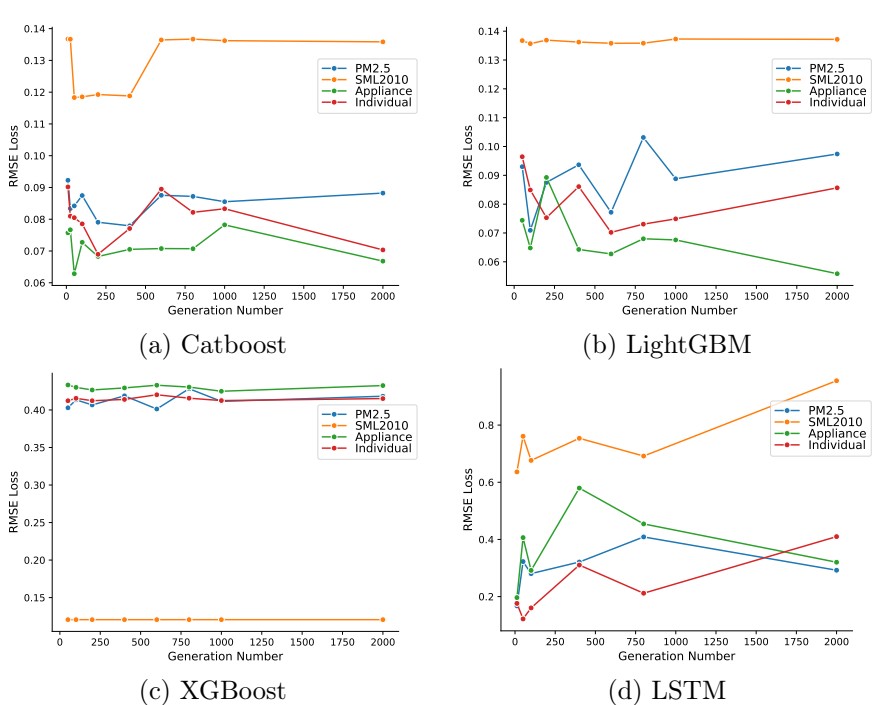

Figure 4: RMSE loss of the final model on the out-of-sample testing set after using the GA to search for hyperparameters. It shows the results after GA runs for different number of generations.

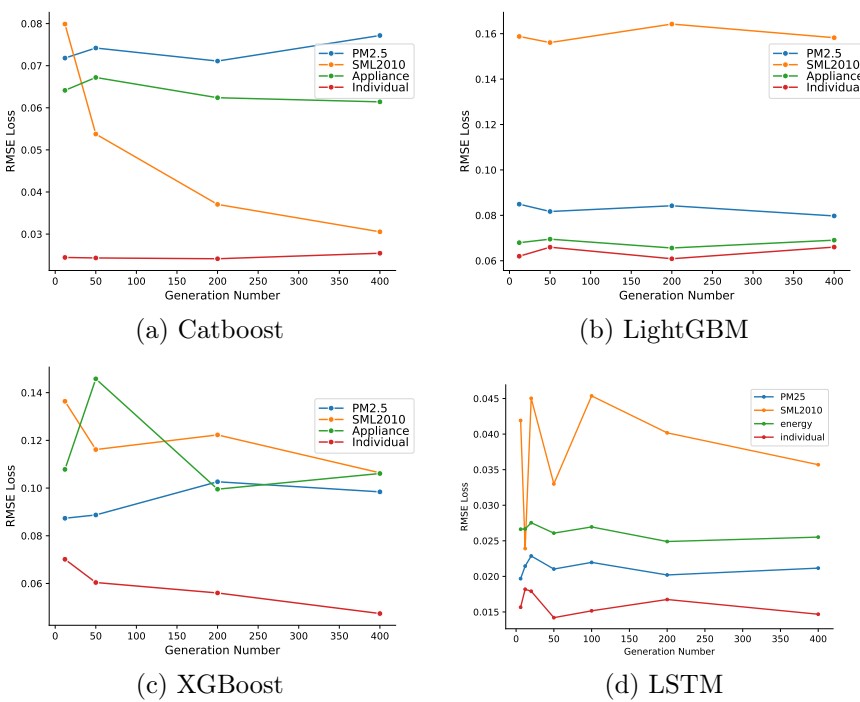

Figure 5: RMSE loss of the final model on the out-of-sample testing set after using SWGA to search for hyperparameters. It shows the results after SWGA runs for different numbers of generations.

