# OpenReview forum: "SWGA: A Distributed Hyperparameter Search Method for Time Series Prediction Models"
_ICLR.cc/2025/Conference — Submitted to ICLR 2025_

### Official Review · Reviewer_fkEy · 2024-10-21

**Soundness:** 1
**Presentation:** 2
**Contribution:** 1
**Rating:** 1
**Confidence:** 3

**Summary:**

This paper introduces a novel hyperparameter scheme based on genetic algorithms and Bayesian optimization. The proposed approach addresses the distribution shift induced by the nature of time series with a sliding window approach.  The approach is applied to time-series prediction on multiple datasets and various models.

**Strengths:**

- the paper is clear and concise, though it presents some nits reported below
 - the method is applied on multiple datasets and on a variety of different models.

**Weaknesses:**

Important weaknesses:
 - the contributions are strictly practical, thus it is of fundamental importance that the method is accessible to researchers and reproducible. Particularly, ICRL strongly encourage to add a “Reproducibility Statement” (https://iclr.cc/Conferences/2025/AuthorGuide), and the paper is missing it. Furthermore, no code is provided as supplementary material, making the results hard to reproduce and inaccessible
 - table 1 shows that it’s the TPE addition that outperforms the other method, but no experiment shows that the proposed method is better than just TPE.
 - results are not reported with confidence intervals. Thus, there is no way to check if the results are just due to variance.
 - No information is given on the training procedure of the GA counterpart. Furthermore, multiple GA algorithms are present in the literature, but it’s never reported which is the one that is applied for the results.
Minor weaknesses
 - line 178, TPE is criticized for its ability to scale badly with dimensionality, though the paper uses it to initialize the population, inheriting its downside
 - the 4th contribution point is pointless, given that no code is provided as supplementary material
 - the 2nd contribution point states that the proposed approach should address the distributions shift induced by the nature of time-series. However, nowhere in the paper this is investigated, and specifically, nowhere is addressed what are the other options apart from SWGA and why they should fall short in time series predictions
 - in the “conclusion” section, it is stated “Additionally, we also demonstrate the good scalability of SWGA”. However, no reference to other algorithms/approaches is given. Thus, with a lack of baselines, it’s completely irrelevant

My recommendation is to reject the paper.
The main reasons behind this opinion are:
 - the contributions are minors, and they can be summarized in a smart initialization of the GA using TPE, and a sliding window training approach.
 - the lack of reproducibility of the results
 - the narrow comparison with other approaches.
———————————————————————————
Writing concerns:
 - citations should be between parenthesis if not part of the main text (e.g. line 39-40 and then the rest of the paper, respectively use \citep for parenthesis and cite for normal citation)
 - line 68, TPE was never defined (will be defined later, though it should be defined on the first usage)
 - “But, their full” line 107
 - “K-fold cross-validation effectively reduces the risk of overfitting”: how? It’s a validation method, not a regularization
 - The equation on line 140 is not numbered, and it contains 2 “i” indexes. Please use different letters to avoid confusion
 - line 161 “while domain adaptation is to”
 - line 258 263, the pseudocode contains a line break that is irrelevant
Personal opinions:
 - contribution 2 and 3 are not contributions, but positive aspects of the method, maybe better to write them outside of the bullet points
 - the first two paragraphs in 4.1 are repetitive, and the explanation on how the GA works should be part of the background section

**Questions:**

I don't have any questions for the authors

---

### Official Review · Reviewer_JPPQ · 2024-10-25

**Soundness:** 1
**Presentation:** 1
**Contribution:** 1
**Rating:** 1
**Confidence:** 4

**Summary:**

The authors propose a generic algorithm (GA) based HPO method that aims to find good hyperparameters assuming the distribution of observations change over time. The way the proposed method works is to sequentially go through the data, as a sliding window (SWGA), and use the springs and top performing configurations as the starting population for the next iteration. The authors also propose to use TPE instead of Random Search to fill the initial population.

**Strengths:**

* It’s a relevant problem for time series forecasting.
* It shows some potential for the sliding window strategy.

**Weaknesses:**

- The contribution is minor. I can’t agree that using TPE instead of Random Search is an actual contribution. To fill the initial population, doing a small HPO with TPE should have better results than random configurations. It’s too obvious. Also, to show the sliding window is a better HPO strategy, it needs to first deliver better quality and second is cost effective. While the authors’ experiments (Table 1 & 2) show SWGA has better quality than GA, I am not sure the baseline of GA is properly implemented due to lacking of detail. Also, there are tons of HPO methods that can be combined with the sliding window and they can be used to show sliding window is indeed helpful.

- The technical correctness is hard to access given the current state. Many important details are missing. For example:
  - How do the authors ensure that the only baseline GA is comparable with the SWGA? For me, to see if the sliding window helps, taking Figure 1 as an example, it should be a single GA on the average performance of 12 splits. Assume population size is M, then SWGA takes trains M*12 times and the single GA baseline also trains M*12 times.
   - Line 333: “we use seven historical timesteps to predict one timestep ahead” Does this mean the authors sample segments of size 8 from training and validation set? How many are sampled?

- The experiments, especially Table 3, does not support the contribution.Table 3 only shows HPO helps, not why sliding window or warm start is a good strategy. The part of scalability does not fit into the current story. The contribution of the paper, as claimed by the authors, are the sliding window and warm up. The focus is not distributed training or scheduling etc.

- The terminology and notations are not precise and conventional. For example:
  - Line 35 ”the model can achieve better performance on out-of-sample data with a matching distribution” What does out-of-sample data with a matching distribution mean? It’s also strange that the authors call the prediction window as out-of-sample data. It may or may not be out-of-sample.
  - Line 141, please check notation, many misusages.

**Questions:**

See the weakness section.

---

### Official Review · Reviewer_tB6e · 2024-10-29

**Soundness:** 2
**Presentation:** 2
**Contribution:** 1
**Rating:** 3
**Confidence:** 4

**Summary:**

This paper introduces a hyperparameter search process specifically tailored for time series forecasting, focusing on temporal distribution shifts in the data. The proposed method is based on a variation of the genetic algorithm and the sliding window validation technique. The algorithm is designed to be parallelizable, which helps further reduce the time needed for hyperparameter search.

**Strengths:**

* The algorithm is parallelizable, significantly decreasing the computation time required to find the optimal set of hyperparameters.
* Evaluation is conducted on a sufficient number of widely known real-life datasets.

**Weaknesses:**

* In the experiments, the proposed algorithm is only compared with the traditional genetic algorithm. It would be beneficial to evaluate other hyperparameter search techniques, such as the classic sliding window search and k-fold validation technique.
* The models evaluated in the experiments lack diversity. There are 3 tree-based models (CatBoost, LightGBM, XGBoost), 1 recurrent model (LSTM), and 1 attention-based model (Transformer).
* The authors mention high computational costs and inefficient exploration of large hyperparameter search spaces as disadvantages of commonly used techniques. However, the experiments only involve accuracy comparisons. Wouldn’t it be beneficial to demonstrate that the proposed hyperparameter search technique converges more quickly?
* In Appendix A.1, search spaces are provided for the DLinear and PatchTST models, which are not included in the experiments.
* The proposed algorithm combines variations of three widely used search techniques from the literature with minimal modifications. The novelty does not seem satisfying.

**Questions:**

* What is the justification for using 3 tree-based models (CatBoost, LightGBM, XGBoost), 1 recurrent model (LSTM), and 1 attention-based model (Transformer) in the experiments? My intuition is that this hyperparameter search would primarily benefit statistical models such as SARIMAX, ETS, etc., which cannot be trained using a traditional k-fold cross-validation approach. At least one of these model types should be included in the experiments.
* Why didn’t you compare the proposed algorithm with commonly used hyperparameter search algorithms in the literature? Do you believe that comparison with the traditional genetic algorithm is sufficient?
* In the experiments, the proposed algorithm is only compared with the traditional genetic algorithm. It would be beneficial to evaluate other hyperparameter search techniques, such as the classic sliding window search and k-fold validation technique.

---

### Official Review · Reviewer_UjSm · 2024-10-30

**Soundness:** 2
**Presentation:** 2
**Contribution:** 1
**Rating:** 3
**Confidence:** 4

**Summary:**

The paper introduces the Sliding Window Genetic Algorithm (SWGA), a distributed hyperparameter optimization method designed for time series prediction models. Key contributions include (1) a configurable sliding window technique that mitigates overfitting from distribution shifts typical in time series data, (2) a warm-up stage employing Bayesian optimization to establish a robust initial population, and (3) compatibility with distributed computing.

**Strengths:**

1. SWGA introduces a combination of genetic algorithms with a sliding window approach tailored specifically for time series forecasting, addressing the distribution shift and non-stationarity challenges unique to this domain.

2. The authors have provided detailed computational procedures by providing multiple algorithm boxes and demonstrated their impact on real-world applications.

**Weaknesses:**

1. The paper writing has too much redundancy that can be simplified or moved to the appendix. It is also not clear why the proposed method is distinguished from other hyper-parameter optimization approaches.

2. In the experiment section, there is also a lack of comparison with other hyperparameter optimization methods specifically designed for time series data.

3. The SWGA algorithm’s performance might be sensitive to parameters like window size, population size, and mutation rates. However, the paper lacks an exploration of how these parameters impact outcomes.

**Questions:**

See weaknesses.

---

### Meta-Review · Area_Chair_2Cn1 · 2024-12-18

**Metareview:**

The paper introduces the Sliding Window Genetic Algorithm (SWGA) for hyperparameter optimization in time series prediction models, aiming to address distribution shifts by using a sliding window, a warm-up stage with Bayesian optimization (TPE), and distributed computing for scalability. However, reviewers raised concerns about the paper's novelty, as the contributions (TPE initialization and sliding window) were seen as minor. The method was mainly compared to traditional genetic algorithms, with insufficient comparisons to other hyper-parameter optimization techniques like k-fold validation. Issues with experimental details, such as missing confidence intervals and unclear notations, were also noted. Additionally, the lack of reproducibility (no code provided) and a narrow focus on tree-based models were highlighted. Overall, I believe this paper needs more work due to its limited contribution and lack of technical rigor.

**Additional Comments On Reviewer Discussion:**

The authors did not respond to the reviewers.

---

### Decision · Program_Chairs · 2025-01-22

Reject